# Guanidine Derivatives of Quinazoline-2,4(1*H*,3*H*)-Dione as NHE-1 Inhibitors and Anti-Inflammatory Agents

**DOI:** 10.3390/life12101647

**Published:** 2022-10-20

**Authors:** Alexander Spasov, Alexander Ozerov, Vadim Kosolapov, Natalia Gurova, Aida Kucheryavenko, Ludmila Naumenko, Denis Babkov, Viktor Sirotenko, Alena Taran, Alexander Borisov, Elena Sokolova, Vladlen Klochkov, Darya Merezhkina, Mikhail Miroshnikov, Nadezhda Ovsyankina, Alexey Smirnov, Yulia Velikorodnaya

**Affiliations:** 1Department of Pharmacology & Bioinformatics, Volgograd State Medical University, 400001 Volgograd, Russia; 2Scientific Center for Innovative Drugs, Volgograd State Medical University, 400087 Volgograd, Russia; 3Department of Pharmaceutical & Toxicological Chemistry, Volgograd State Medical University, 400001 Volgograd, Russia; 4Department of Pathological Anatomy, Volgograd State Medical University, 400131 Volgograd, Russia

**Keywords:** NHE-1, cytokine release, macrophage, quinazoline-2,4(1*H*,3*H*)-dione, lung injury, LPS

## Abstract

Quinazolines are a rich source of bioactive compounds. Previously, we showed NHE-1 inhibitory, anti-inflammatory, antiplatelet, intraocular pressure lowering, and antiglycating activity for a series of quinazoline-2,4(1*H*,3*H*)-diones and quinazoline-4(3*H*)-one guanidine derivatives. In the present work, novel $N^{1}$,$N^{3}$-bis-substituted quinazoline-2,4(1*H*,3*H*)-dione derivatives bearing two guanidine moieties were synthesized and pharmacologically profiled. The most potent NHE-1 inhibitor **3a** also possesses antiplatelet and intraocular-pressure-reducing activity. Compound **4a** inhibits NO synthesis and IL-6 secretion in murine macrophages without immunotoxicity and alleviates neutrophil infiltration, edema, and tissue lesions in a model of LPS-induced acute lung injury. Hence, we considered quinazoline derivative **4a** as a potential agent for suppression of cytokine-mediated inflammatory response and acute lung injury.

## 1. Introduction

The sodium–hydrogen exchanger NHE-1 is a widely expressed membrane protein responsible for maintaining intracellular pH in a variety of cells, including those of the immune system. It was shown that the activity of NHE-1 regulates many functions of immune cells, including migration, cytokine and chemokine release, lysosomal activity, and Ca^2+^ homeostasis [1,2]. Moreover, the NHE-1 inhibitor amiloride was shown to reduce LPS-induced secretion of IL-1β and TNF-α and to stimulate anti-inflammatory IL-10 in alveolar epithelial cells [3,4], which suggests that NHE-1 also plays a role in lung inflammatory response (Figure 1) [5].

The major class of NHE-1 inhibitors is acylated guanidine derivatives, exemplified by amiloride, cariporide, and eniporide [6]. In an attempt to expand the chemical space of NHE-1 inhibitors, we reported $N^{1}$-alkyl quinazoline-2,4(1*H*,3*H*)-diones and quinazoline-4(3*H*)-ones comprising an *N*-acylguanidine or 3-acyl(5-amino-1,2,4-triazole) side chain as NHE-1 inhibitors endowed with antiplatelet activity [7]. Some of them also reduced rat intraocular pressure and suppressed the formation of advanced glycation end-products, as well as LPS-induced activation of macrophages and also exhibited antidepressant activity comparable to amiloride. In an attempt to optimize these compounds and derive more potent anti-inflammatory agents, we designed new derivatives of quinazolin-2,4(1*H*,3*H*)-dione with linear and cyclic guanidine moieties and performed an in-depth study of the pharmacological properties.

## 2. Materials and Methods

### 2.1. Chemistry

#### 2.1.1. General

All reagents were obtained from Panreac and Acros Organics at the highest grade available and used without further purification. Anhydrous DMF was purchased from Sigma-Aldrich, St. Louis, MO, USA. Thin-layer chromatography (TLC) was performed on Merck TLC Silica gel 60 F${}_{254}$ plates by eluting with CHCl${}_{3}$-MeOH (90:1) or ethanol, which was developed with a VL-6.LC UV lamp (Vilber Lourmat Deutschland GmbH, Eberhardzell, Germany). Yields refer to spectroscopically (NMR) homogeneous materials. The melting points were determined in glass capillaries on a Mel-Temp 3.0 apparatus (Barnstead International, Dubuque, IA, USA). The NMR spectra were recorded using a Bruker Avance 600 (600 MHz for ^1^H and 150 MHz for ^13^C) spectrometer in DMSO-d_6_ or D_2_O with tetramethylsilane as an internal standard. HRMS data were acquired on TripleTOF 5600+ Φ (AB Sciex LLC, Framingham, MA, USA), spray voltage 5.5 kV for positive ions, 4.5 kV for negative ions, aux gas flow rate 15 a.u., and sheath gas flow rate 20 a.u., and the samples were introduced at 30 μL/min.

#### 2.1.2. Dibenzyl 2,2’-(2,4-Dioxoquinazoline-1,3(2*H*,4*H*)-diyl)diacetate (**2a**)

A mixture of quinazolin-2,4(1*H*,3*H*)-dione **1a** (2.50 g, 15.4 mmol), benzyl ester of bromoacetic acid (7.00 g, 30.6 mmol), and anhydrous finely ground K_2_CO_3_ (10.00 g, 72.64 mmol) was stirred in a DMF solution (200 mL) at room temperature for 24 h. The reaction mass was filtered, evaporated to dryness in vacuo; treatment with 100 mL of water afforded the solid residue, which was filtered off, air-dried at room temperature, and purified with recrystallized from ethyl acetate. White solid; yield 62%; mp 94–97 °C; R${}_{f}$ 0.73 (CHCl_3_−MeOH, 90:1); ^1^H NMR (DMSO-d_6_, 600 MHz) δ 8.11 (1H, d, *J* = 8 Hz, H-5), 7.77 (1H, t, *J* = 8 Hz, H-7), 7.47 (1H, d, *J* = 8 Hz, H-8), 7.32–7.40 (11H, m, Ph, H-6), 5.22 (2H, s, CH_2_O), 5.20 (2H, s, CH_2_O), 5.10 (2H, s, NCH_2_C(O)), 4.83 (2H, s, CH_2_O); ^13^C NMR (DMSO-d_6_, 150 MHz) δ 167.8, 167.6, 160.6, 150.2, 139.6, 135.8, 135.6, 135.5, 128.4, 128.1, 128.1, 127.9, 127.8, 123.5, 114.7, 114.3, 66.6, 66.4, 44.9, 42.5; HRMS-ESI: MH^+^, found: C_26_H_23_N_2_O_6_ [M + H]^+^ 459.1552, requires: 459.1551.

#### 2.1.3. Dibenzyl 2,2’-(6-Bromo-2,4-dioxoquinazoline-1,3(2*H*,4*H*)-diyl)diacetate (**2b**)

Compound **2b** was synthesized similarly from 6-bromoquinazolin-2,4(1*H*,3*H*)-dione (**1b**). Light yellow solid; yield 65%; mp 124–127 °C; R${}_{f}$ 0.79 (CHCl_3_−MeOH, 90:1); ^1^H NMR (DMSO-d_6_, 600 MHz) δ 8.15 (1H, d, *J* = 2.5 Hz, H-5), 7.94 (1H, d, *J* = 9 Hz, H-7), 7.49 (1H, d, *J* = 9 Hz, H-8), 7.32–7.39 (10H, m, Ph), 5.21 (2H, s, CH_2_O), 5.20 (2H, s, CH_2_O), 5.09 (2H, s, NCH_2_C(O)), 4.81 (2H, s, CH_2_O); ^13^C NMR (DMSO-d_6_, 150 MHz) δ 167.6, 167.4, 159.5, 149.9, 138.9, 138.3, 135.5, 135.4, 129.9, 128.4, 128.2, 128.2, 127.9, 127.9, 117.5, 116.0, 115.5, 66.7, 66.5, 45.1, 42.6; HRMS-ESI: MH^+^, found: C_26_H_22_BrN_2_O_6_ [M + H]^+^ 537.0646, requires: 537.0656.

#### 2.1.4. 2,2’-(2,4-Dioxoquinazoline-1,3(2*H*,4*H*)-diyl)bis(N-carbamimidoylacetamide) (**3a**)

A mixture of **2a** (2.30 g, 5.02 mmol), guanidine hydrochloride (1.00 g, 10.5 mmol), and KOH (0.60 g, 10.7 mmol) was refluxed in 95% ethanol solution (50 mL) for 10 min. The hot reaction mass was filtered and cooled. The solid residue was filtered off, dried at room temperature, and twice recrystallized from ethanol. White solid; yield 81%; mp 336–338 °C; R${}_{f}$ 0.59 (95% EtOH; ^1^H NMR (D_2_O 600 MHz) δ 8.10 (1H, d, *J* = 8 Hz, H-5), 7.81 (1H, t, *J* = 8 Hz, H-7), 7.38 (1H, t, *J* = 8 Hz, H-6), 7.20 (1H, d, *J* = 8 Hz, H-8), 4.58 (2H, s, NCH_2_C(O)), 4.57 (2H, s, NCH_2_C(O)); ^13^C NMR (D_2_O, 150 MHz) δ 174.5, 174.3, 162.7, 157.5, 151.1, 139.4, 135.8, 127.7, 123.5, 114.2, 113.9, 46.9, 44.7; HRMS-ESI: MH^+^, found: C_14_H_17_N_8_O_4_ [M + H]^+^ 361.1380, requires: 361.1373.

#### 2.1.5. 2,2’-(6-Bromo-2,4-dioxoquinazoline-1,3(2*H*,4*H*)-diyl)bis(N-carbamimidoylacetamide) (**3b**)

Compound **3b** was synthesized similarly from **2b**. White solid; yield 68%; mp >400 °C; R${}_{f}$ 0.33 (95% EtOH; ^1^H NMR (D_2_O, 600 MHz) δ 8.17 (1H, d, *J* = 2.5 Hz, H-5), 7.90 (1H, d *J* = 8 Hz, H-7), 7.15 (1H, d, *J* = 8 Hz, H-8), 4.65 (2H, s, NCH_2_C(O)), 4.54 (2H, s, NCH_2_C(O)); ^13^C NMR (D_2_O, 150 MHz) δ 173.2, 172.9, 161.0, 157.5, 150.6, 138.9, 138.0, 129.8, 116.4, 116.1, 115.4, 47.0, 44.9; HRMS-ESI: MH^+^, found: C_14_H_16_BrN_8_O_4_ [M + H]^+^ 439.0470, requires: 439.0478.

#### 2.1.6. 1,3-Bis[(5-amino-4H-1,2,4-triazol-3-yl)methyl]quinazoline-2,4(1*H*,3*H*)-dione (**4a**)

A mixture of **2a** (2.30 g, 5.02 mmol), aminoguanidine carbonate (1.45 g, 10.7 mmol), and KOH (1.20 g, 21.4 mmol) was refluxed in 95% ethanol solution (50 mL) for 1 h. The hot reaction mass was filtered and cooled. The solid residue was filtered off, dried at room temperature, and twice recrystallized from ethanol. White solid; yield 63%; mp 314–318 °C; R${}_{f}$ 0.69 (80% EtOH; ^1^H NMR (D_2_O, 600 MHz) δ 10.14 (1H, d, *J* = 8 Hz, H-5), 9.85 (1H, t, *J* = 8 Hz, H-7), 9.42 (1H, t, *J* = 8 Hz, H-6), 9.25 (1H, d, *J* = 8 Hz, H-8), 6.64 (2H, s, CH_2_), 6.52 (2H, s, CH_2_); ^13^C NMR (D_2_O, 150 MHz) δ 175.0, 174.8, 164.1, 162.0, 152.9, 141.9, 137.6, 129.8, 125.2, 116.5, 116.2, 49.0, 46.9; HRMS-ESI: MH^+^, found: C_14_H_15_N_10_O_2_ [M + H]^+^ 355.1382, requires: 355.1379.

#### 2.1.7. 1,3-Bis[(5-amino-4H-1,2,4-triazol-3-yl)methyl]-6-bromoquinazoline-2,4(1*H*,3*H*)-dione (**4b**)

Compound **4b** was synthesized similarly from **2b**. White solid; yield 60%; mp >400 °C; R${}_{f}$ 0.67 (80% EtOH; ^1^H NMR (D_2_O, 600 MHz) δ 8.18 (1H, s, H-5), 7.87 (1H, d *J* = 8 Hz, H-7), 7.12 (1H, d, *J* = 8 Hz, H-8), 4.73 (2H, s, CH_2_), 4.62 (2H, s, CH_2_); ^13^C NMR (D_2_O, 150 MHz) δ 174.1, 173.9, 161.5, 150.8, 138.5, 138.2, 129.9, 116.0, 115.8, 115.7, 47.0, 44.8; HRMS-ESI: MH^+^, found: C_14_H_14_BrN_10_O_2_ [M + H]^+^ 433.0479, requires: 433.0485.

### 2.2. Cellular Assays

#### 2.2.1. NHE-1 Inhibition Assay

Inhibition of rabbit platelet NHE-1 was determined according to the method of [8,9] in our modification, which was reported by us previously [7] using the laser aggregometer BIOLA-220 LA (Russia).

#### 2.2.2. Platelet Aggregation Assay

Platelet aggregation was assessed on a two-channel laser analyzer “BIOLA-220 LA” (Russia) as described previously [10] using rabbit platelets.

#### 2.2.3. Isolation and Stimulation of Primary Macrophages

Primary C57bl/6j murine macrophages were elicited with intraperitoneal injection of 3% peptone solution as per the standard procedure [11]. Stimulation of inflammatory response was performed with *E. coli* O127:B8 LPS (100 ng/mL final concentration).

#### 2.2.4. Assay of Nitric Oxide and Cytokines

Nitric oxide was determined in culture supernatants as a nitrite anion with a standard Griess reagent after 22 h of incubation with test compounds at a wavelength of 550 nm with a microplate reader Infinite M200 PRO (Tecan GmbH, Grodig, Austria). Concentrations of IL-6 and TNF-α in cell supernatants were quantified with commercial ELISA kits according to the manufacturer’s instructions (Cloud-clone Corp., Katy, TX, USA).

#### 2.2.5. Cytotoxicity Study

The activity of lactate dehydrogenase (LDH) in a cell culture medium was determined as reported by us previously [7] with a microplate reader Infinite M200 PRO (Tecan, Austria).

#### 2.2.6. Phagocytosis Assay

The phagocytic activity of peritoneal macrophages was assessed after staining with Azur-Eosin with Romanovsky’s modifications as reported by us previously [7] with a light microscopy using a Mikmed-6 (LOMO, Saint Petersburg, Russia) equipped with a digital camera.

### 2.3. Animal Studies

The reported study adhered to the ARRIVE Guidelines [12]. Male C57bl/6j mice (21–24 g) were housed 5 per cage in ambient lighting and 60% humidity. Animals were provided with free access to water and standard chow prior to the study.

#### 2.3.1. LPS-Induced Acute Lung Injury

Prior to the experiment, C57BL/6J mice were randomized according to body weight and motor activity in an open field test. Reference drug dexamethasone (5 mg/kg) or tested compound **4a** (30 mg/kg) was administered to the respective experimental groups with intraperitoneal injection in 10 mL/kg of sterile saline. Animals of the control group were injected with an equal volume of the vehicle. Animals were anesthetized with isoflurane inhalation until the breathing rate decreased 1 h later. Mice were suspended by the front incisors on an inclined surgical table; the tongue was pulled out with narrow curved tweezers, and 1 mg/mL of *E. coli* O127:B8 LPS (Sigma-Aldrich, Israel) in 1 mL/kg sterile saline was injected into the back of the oropharynx to allow aspiration [13]. Intact animals received an equal volume of sterile saline in a similar way.

#### 2.3.2. Open Field Test

The open field test was performed with the NPO “Open Science” (Russia) arena of 44 cm diameter and 32 cm wall height under 300 lux lighting as reported by us previously [7].

#### 2.3.3. Bronchoalveolar Lavage and Blood Plasma Preparation

Mice were anesthetized with 500 mg/kg chloral hydrate (Sigma-Aldrich, Germany) intraperitoneally 24 h after LPS administration. Blood and bronchoalveolar lavage (BAL) sampling was performed as described in [13]. Leukocyte counts and the lung permeability index were determined as described by us previously [7].

#### 2.3.4. Intraocular Pressure Study

Intraocular pressure was measured on 75 adult outbred rats of both sexes with a TonoVet device (Finland) as reported by us previously [7].

#### 2.3.5. Tail Suspension Test

Antidepressant activity was determined on 36 male ICR mice weighing 22–25 g as described in [14] using the Panlab LE808 apparatus.

#### 2.3.6. Histological Study

Histological and immunohistological assessment of tissue sections was performed in a semi-quantitative way [15] on paraffin sections after hematoxylin and eosin staining with a light microscope (Zeiss, Germany) with double-blinding. The degree of inflammation was determined as previously reported [7].

### 2.4. Data Analysis

Statistical analysis was performed in Prism 8.0 (GraphPad Software, San Diego, CA, USA). The nonparametric Mann–Whitney U-test was used for pairwise comparisons and 1-way ANOVA with a Dunnett post-test for multiple comparisons. IC${}_{50}$ values were obtained with a nonlinear 3-parametric regression.

## 3. Results

### 3.1. Chemistry

The target compounds were designed and realized to comprise quinazoline-2,4(1*H*,3*H*)-dione bis-substituted at nitrogen atoms with either linear guanidine (**3a**, **3b**) or cyclic guanidine analogue 5-amino-1,2,4-triazole (**4a**, **4b**). Bromine substitution of H_5_ was also realized to evaluate higher lipophilicity and steric limitations.

The synthetic route is shown in Figure 2. Starting quinazoline-2,4(1*H*,3*H*)-dione **1a** and 6-bromoquinazoline-2,4(1*H*,3*H*)-dione **1b** were readily alkylated with 2 molar equivalents of benzyl bromoacetate at ambient temperature in anhydrous DMF using an excess of potassium carbonate as a base. The resulting esters **2a**,**b** were obtained in a 62 and 65% yield, respectively. Next, esters **2a**,**b** were treated with guanidine generated *in situ* from 2 molar equivalents of guanidine hydrochloride and potassium hydroxide in boiling 95% ethanol, which led to rapid cleavage of the ester bond and the formation of *N*-acyl derivatives of guanidine **3a**,**b** in an 81 and 68% yield.

When aminoguanidine was similarly obtained as a nucleophilic reagent in situ from aminoguanidine carbonate and potassium hydroxide in boiling 95% ethanol, the reaction was accompanied by cyclization to form 5-amino-1,2,4-triazole and led to quinazoline-2,4(1*H*,3*H*)-dione derivatives **4a**,**b** with 63 and 60% yields.

### 3.2. NHE-1 Inhibition

Target compounds were evaluated as inhibitors of rabbit platelet NHE-1 (Table 1). It was shown that the substitution of $H^{5}$ with bromine decreased the activity of compounds **3b** and **4b**, indicating possible steric limitations of the binding site. In turn, guanidine derivative **3a** is the most active NHE-1 inhibitor with IC${}_{50}$ in the nanomolar range. Its 5-amino-1,2,4-triazole counterpart **4a** also demonstrated high potency exceeding amiloride by four times.

### 3.3. Antiplatelet Activity

The target compounds markedly inhibited ADP-induced platelet aggregation. At 100 μM, they demonstrated comparable efficiency, exceeding the reference drug acetylsalicylic acid, but were inferior to amiloride.

### 3.4. IOP-Lowering Activity

To access the influence on rat eye aqueous humor formation, the studied compounds were administered as 0.4% eye drops. The selective NHE-1 inhibitor zoniporide exhibited the highest IOP reduction. Target derivatives demonstrated various degrees of intraocular pressure reduction with compound **4a** being the most active. Introduction of 5-bromine (**4b**) or acyclic guanidine $N^{1}$,$N^{3}$-side chains (**3a**) led to 2- and 3-fold activity reduction, respectively.

### 3.5. Antidepressant Activity

The tail suspension test was used to assess the antidepressant activity of the most-active NHE-1 inhibitors. Derivatives **4a** and **4b** demonstrated antidepressant activity comparable to amiloride [16], but inferior to imipramine or amitriptyline (Table 2).

### 3.6. Anti-Inflammatory Activity

Given our primary focus on the development of agents against cytokine-mediated tissue damage, we evaluated the target compounds as inhibitors of pro-inflammatory macrophage activation. Primary peritoneal macrophages were isolated from C57bl/6j mice, pooled, and stimulated with *E. coli* LPS as TLR4 agonist. Pro-inflammatory activation was assessed as nitric oxide (NO) synthesis and interleukin-6 (IL-6) secretion. The cytotoxicity of the compounds was monitored in parallel using the lactate dehydrogenase assay.

We found that compounds **4a** and **4b** effectively inhibited LPS-induced NO synthesis at high micromolar concentrations, while their counterparts **3a** and **3b**, comprising acyclic guanidine side chains, were essentially inactive (Table 3). Compound **4a** was the most active as an inhibitor of IL-6 secretion with IC${}_{50}$ of 51.75 μM. Substitution of H^5^ on bromine had little impact on the anti-inflammatory activity. Reference drug amiloride also inhibited IL-6 secretion in the micromolar range. Given the potent anti-inflammatory activity of **4a**, it was further evaluated as a lead compound.

### 3.7. Influence of Compound ***4a*** on Macrophage Phagocytic Activity

In an attempt to identify safer alternatives to steroid anti-inflammatory agents, it is mandatory to evaluate lead compounds for immunodepressant activity. To do so, we determined the influence of compound **4a** on macrophage phagocytosis after 72 h of incubation. Cell viability was monitored in parallel via the LDH assay.

The dexamethasone-treated cells demonstrated a sharp drop in phagocytic activity (41% from control), as expected [17,18] (Figure 3). The mean number of phagocytosed particles per cell was 59%. Lead compound **4a** insignificantly influenced macrophage phagocytic activity, but diminished the mean number of phagocytosed particles by 23%. Dexamethasone did not influence cell viability, while **4a** slightly decreased it. Macrophage spreading was registered and visually scored as a phenotypic marker of cell activation phagocytic capacity [15,19]. Macrophages treated with yeast were partially stimulated, while compound **4a** marginally reduced spreading. The effect of dexamethasone was more evident. Thus, dexamethasone had a marked immunosuppressive effect, but compound **4a** demonstrated a negligible influence on the innate phagocytic activity of macrophages.

### 3.8. Protective Activity of ***4a*** in Murine Model of Acute Lung Injury

To assess the anti-inflammatory activity of the lead compound **4a** in vivo, we used the LPS-induced acute lung injury model. After randomization, C57bl/6j mice were treated with 50 mg/kg **4a**, 5 mg/kg dexamethasone, or the vehicle, followed by intratracheal instillation of 5 mg/kg *E. coli* LPS. Lung injury was assessed 24 h later based on behavioral, biochemical, cytological, and morphological alterations.

The behavior of the animals was assessed using an open field test twice, 2 and 22 h after LPS treatment. Decreased motor, exploratory, and orienting activity of control animals persisted throughout the experiment. Dexamethasone reduced the LPS-induced motor and behavioral alterations. A similar effect was observed for dexamethasone previously [17,18]. Compound **4a** also restored the motor activity of animals, although the effect size was smaller than in the dexamethasone group (statistically insignificant). Both dexamethasone and **4a** comparably improved exploratory activity (Figure 4).

LPS treatment elicited acute lung inflammation (Figure 5). Accumulation of IL-6 in alveolar lumen and blood was markedly increased. At the same time, TNF-α levels were slightly reduced, which is consistent with the literature data [20,21]. Increased alveolar vascular permeability for blood plasma proteins supported the development of acute lung injury (Figure 2). Dexamethasone and compound **4a** restored the markers of inflammation, e.g., significantly reduced plasma IL-6. Both dexamethasone and **4a** preserved pulmonary vessels’ permeability, thus preventing the development of pulmonary edema (Figure 6. The concentration of TNF-α both in bronchoalveolar liquid (BAL) and in blood plasma turned out to be a non-informative marker, since its secretion peaked at 3–6 h after LPS administration (lit, data), and after 24 h after LPS administration, its level did not differ from the intact group. In summary, dexamethasone and **4a** prevented cytokine release and preserved vascular integrity.

Examination of the cellular composition of BAL after LPS administration revealed a drastic prevalence of mature segmented neutrophils as compared to the intact animals (*p* < 0.05) accompanied by the depletion of monocytes, reflecting the development of an acute inflammatory reaction (Figure 7). Treatment with dexamethasone or **4a** significantly ameliorated the accumulation of segmented neutrophils and restored the monocyte content in BAL. It is noteworthy that **4a** showed the most pronounced inhibition of the neutrophil migration to the site of inflammation.

Similar trends were observed in the leukocyte formula (Figure 7). Dexamethasone confirmed its potent anti-inflammatory activity by preserving the proportion of mature segmented neutrophils compared to the LPS-treated group. Compound **4a** also restored the ratio of segmented neutrophils to lymphocytes. Other leukocyte sub-populations showed no pathological changes in comparison to the intact animals.

Histological examination of the lung tissue of LPS-treated animals revealed a significant infiltration of interstitial lung tissue by polymorphonuclear neutrophils along with purulent exudate in the alveolar space and bronchioles (Figure 8). Dexamethasone significantly reduced these hallmarks of inflammation. Specifically, the inflammatory infiltrate was represented mainly by alveolar macrophages and macrophage-like cells, while thickening of the alveolar septa and interstitial edema due to serous and serous-hemorrhagic exudative inflammation were less pronounced. Compound **4a** reduced LPS-induced inflammation in mouse lung tissue. Focal infiltration of the interstitial lung tissue by neutrophilic leukocytes was noted only in two experimental animals of the group. It is noteworthy that this was not accompanied by purulent exudate into the lumen of the alveoli. Hence, compound **4a** preserved the normal structures of the lung tissue, especially the alveolar septa. The slight thickening of the latter and interstitial edema were mainly due to exudative inflammation of a serous-hemorrhagic nature.

Mononuclear phagocytes were identified using immunohistochemical staining of CD68 glycoprotein [22]. In the lung tissue of vehicle-treated animals, CD68${}^{+}$ cells were found in the adventitial membrane of bronchioles (Figure 9), on the surface of the alveoli, and occasionally, in the interalveolar septa and lung interstitium. LPS administration led to the accumulation of large CD68${}^{+}$ macrophages in the interstitium infiltrate, often in the thickened walls and on the surface of the alveoli. Dexamethasone significantly reduced this phenomenon; CD68${}^{+}$ macrophages were rarely detected in the cellular infiltrate. Single CD68${}^{+}$ cells were also found on the alveolar surface. Compound **4a** effectively limited LPS-induced inflammation. Single CD68${}^{+}$ cells were found in the infiltrate, usually on the surface and in the walls of the alveoli, making the distribution of CD68${}^{+}$ macrophages virtually identical to intact animals.

## 4. Discussion

Quinazoline is a versatile scaffold for the construction of molecules with a wide range of biological activities, including anti-inflammatory agents [23], e.g., there are quinazoline derivatives reported to be active in animal models of adjuvant arthritis [24], carrageenin-induced edema [25], formaldehyde-induced edema [26], and LPS-induced lung injury [27]. It is noteworthy that, in the majority of these studies, researchers explored *N*-substituted 4-(arylamino)quinazolines or 2,4-disubstituted compounds.

In turn, our data suggest that the introduction of the guanidine moiety to 1,3-disubstituted quinazolin-2,4(1*H*,3*H*)-diones is a promising approach for the development of NHE-1 inhibitors. In line with previously reported SAR observations [7], 5-amino-1,2,4-triazole containing derivatives proved to be more active, suggesting that the substitution of the guanidine residue with the conformationally rigid 5-amino-1,2,4-triazole guanidine “mimic” is favorable for NHE-1 inhibition.

Additionally, we tested the target compounds for antiplatelet and intraocular-pressure-reducing activity, which can be attributed, at least in part, to NHE-1 inhibition [28,29]. While all studied compounds demonstrated comparable platelet aggregation inhibition, IOP reduction showed non-additive SAR, i.e., the introduction of 5-bromine and substitution of the guanidine moiety with 5-amino-1,2,4-triazole improved the activity, while their combination showed inferior results. These discrepancies and the lack of correlation with NHE-1 inhibitory activity may be attributed to the engagement of targets other than NHE-1 or differences in the eye tissue barriers’ permeability.

Considering the pivotal role that NHE-1 plays in maintaining immune cell response to LPS [2,30], we thoroughly assessed the anti-inflammatory activity of the target compounds. It was found that triazole-containing analogues **4a** and **4b** dose-dependently inhibited NO synthesis and IL-6 secretion in primary murine macrophages without apparent cytotoxicity. The anti-inflammatory effect of **4a** and **4b** was indirectly confirmed by antidepressant activity *in vivo*, since microglia activation is dependent on NHE-1 activity [31]. The most active compound **4a** had a minimal impact on macrophage phagocytic function in the 72 h study. Hence, lead compound **4a** inhibited NHE-1 and LPS-induced IL-6 secretion, preserving innate immunity, which is an important advantage compared to steroid anti-inflammatory drugs.

Clinical data suggest that cytokine release governs and perpetuates acute lung injury, amplifying immune-mediated tissue damage. Given that the NHE-1 inhibitor amiloride attenuates LPS-induced acute lung injury in rats, we evaluated compound **4a** as a protective agent after intratracheal LPS administration to C57bl/6j mice. We found that treatment with **4a** ameliorated behavioral defects associated with acute inflammation, limited IL-6 secretion, preserved normal alveolar vascular permeability, and prevented the migration of neutrophils into the alveoli lumen. Moreover, interstitial edema, hemorrhage, and macrophage infiltration were effectively prevented by **4a**. Taken together, our findings suggest that quinazolin-2,4(1*H*,3*H*)-diones bearing 5-amino-1,2,4-triazole side chains open a venue for the development of safer and more-efficient agents inflammation-mediated lung injury.

## Figures and Tables

**Figure 1 life-12-01647-f001:**
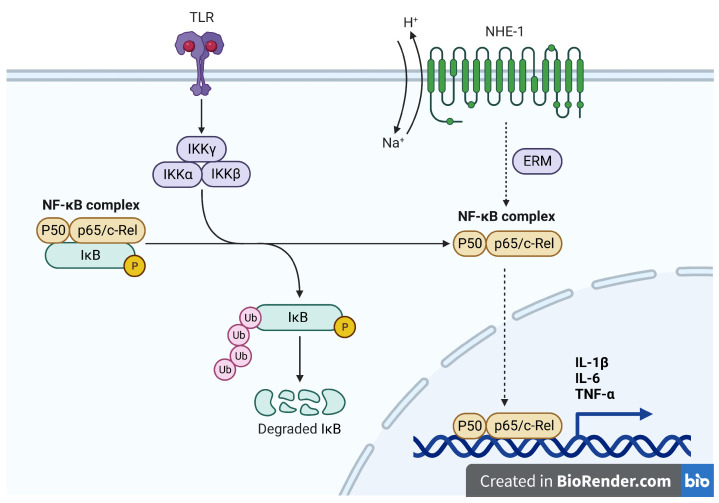
Putative role of NHE-1 in inflammation.

**Figure 2 life-12-01647-f002:**
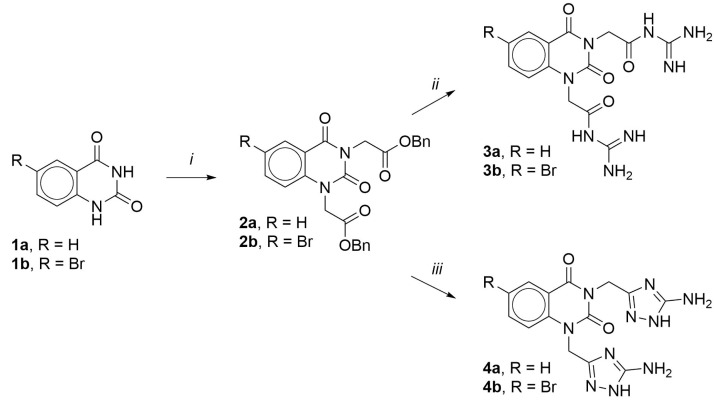
Synthesis of the target compounds: (i) BrCH_2_C(O)OBn, K_2_CO_3_, DMF, 25 °C, 24 h, 62–65%; (ii) NH_2_C(NH)NH_2_ · HCl, KOH, 95% EtOH, reflux, 10 min, 68–81%; (iii) NH_2_C(NH)NHNH_2_ ·$\frac{1}{2}$H_2_CO_3_, KOH, 95% EtOH, reflux, 1 h, 60–63%.

**Figure 3 life-12-01647-f003:**
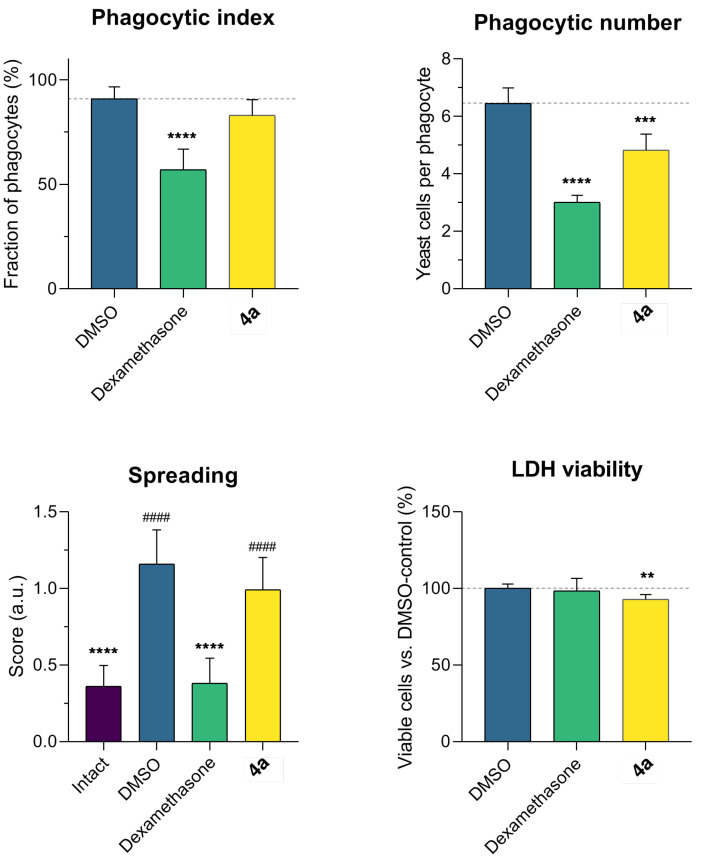
Compound **4a** has minimal impact on phagocytic activity and viability of C57bl/6j peritoneal macrophages. Data as the mean and 95% C.I. (n = 100). Statistical significance: ** *p* < 0.01, *** *p* < 0.001, **** *p* < 0.0001 vs. DMSO; ^####^ *p* < 0.0001 vs. intact cells (1-way ANOVA, Dunnet’s post-test).

**Figure 4 life-12-01647-f004:**
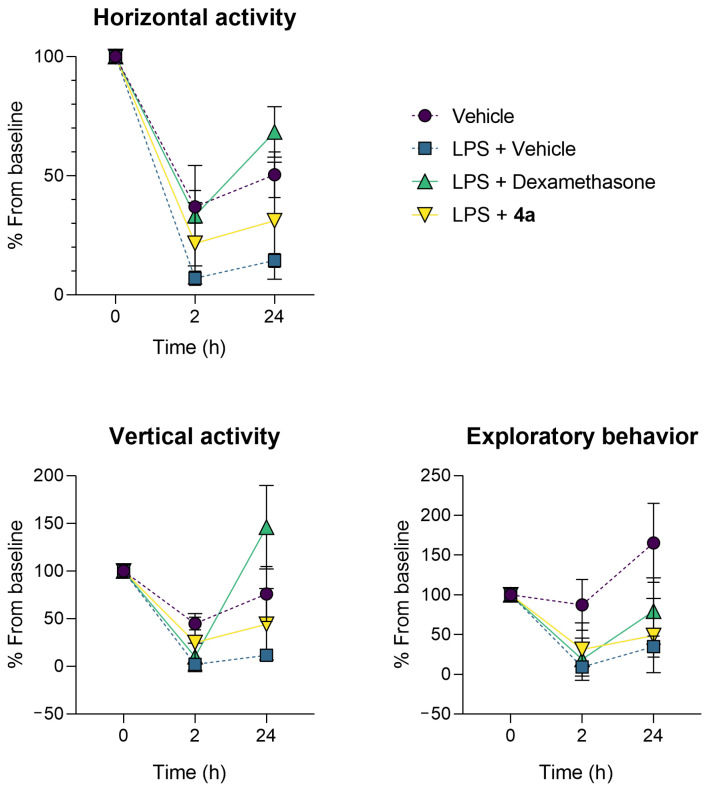
Compound **4a** alleviated sickness behavior in the C57bl/6j murine model of LPS-induced acute lung injury. Data as the mean and SD, n = 5.

**Figure 5 life-12-01647-f005:**
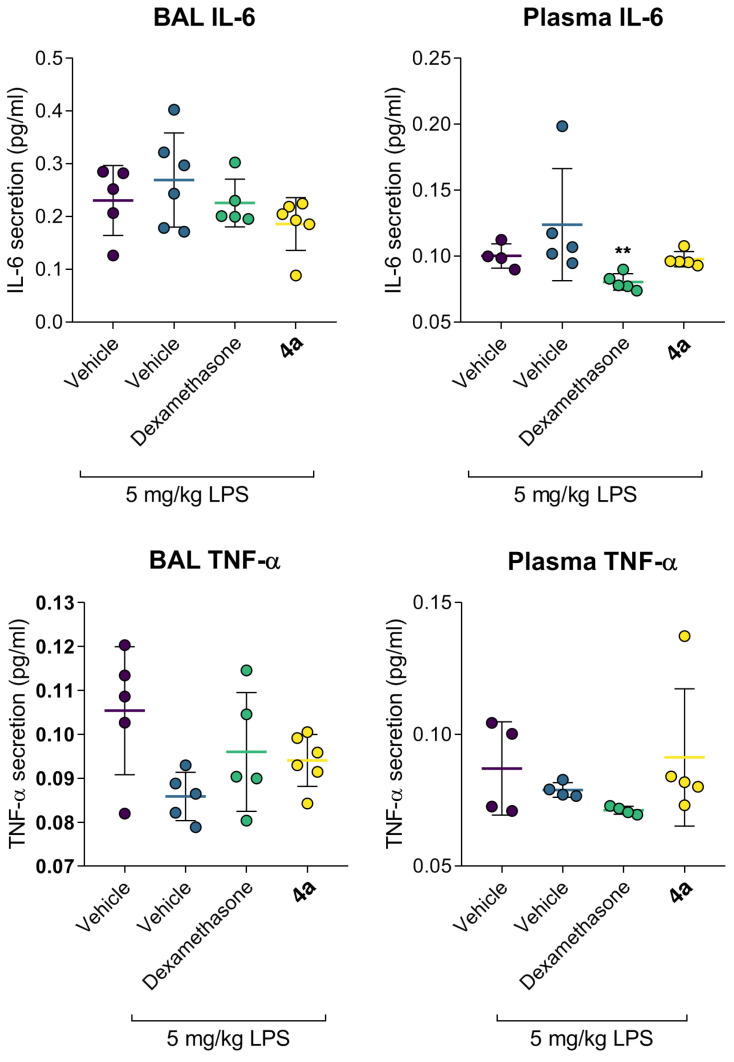
Compound **4a** limits IL-6 secretion in the C57bl/6j murine model of LPS-induced acute lung injury. Data as the mean and SD, n = 5. Statistical significance vs. LPS (1-way ANOVA, Dunnet’s post-test): ** *p* < 0.01.

**Figure 6 life-12-01647-f006:**
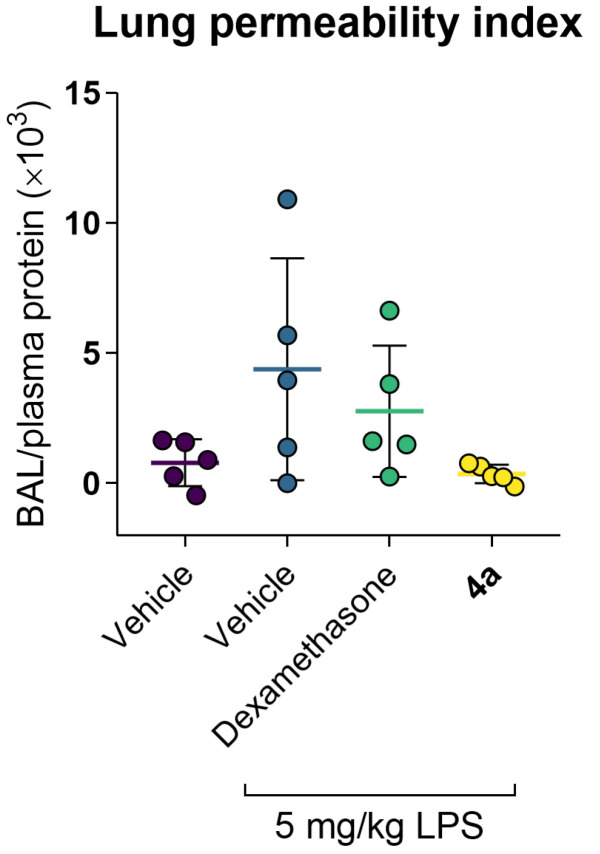
Compound **4a** preserves normal permeability of alveolar vessels in the C57bl/6j murine model of LPS-induced acute lung injury. Data as the mean and SD, n = 5.

**Figure 7 life-12-01647-f007:**
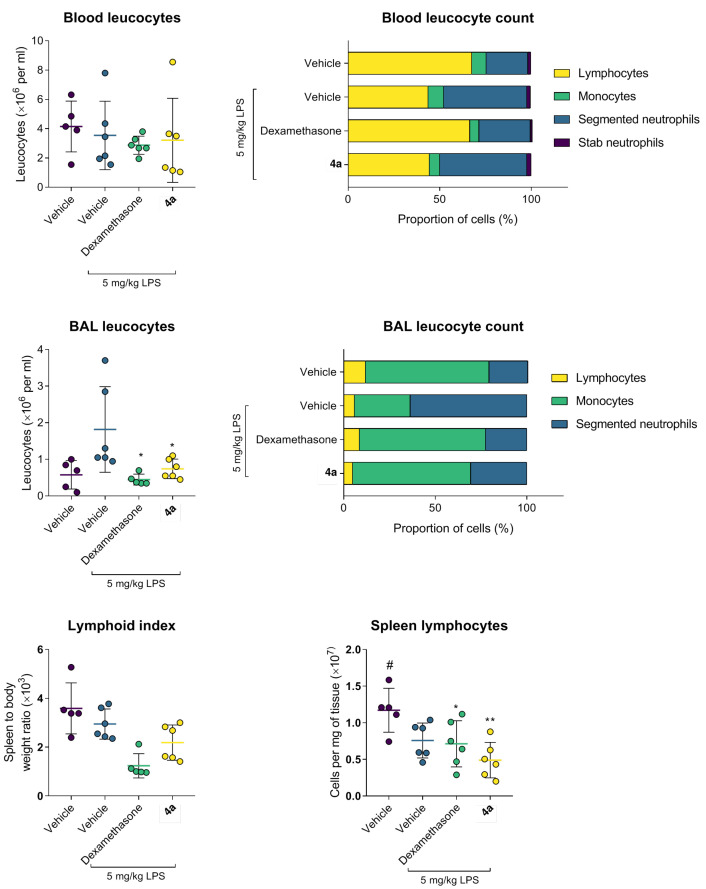
Compound **4a** prevents leukocyte infiltration and lymphocyte recruitment in the C57bl/6j murine model of LPS-induced acute lung injury. Data as the mean and SD, n = 5. Statistical significance (1-way ANOVA, Dunnet’s post-test): * *p* < 0.05 vs. Vehicle + LPS (1-way ANOVA, Dunnet’s post-test); ** *p* < 0.01 vs. LPS; ^#^ *p* < 0.05 vs. Vehicle.

**Figure 8 life-12-01647-f008:**
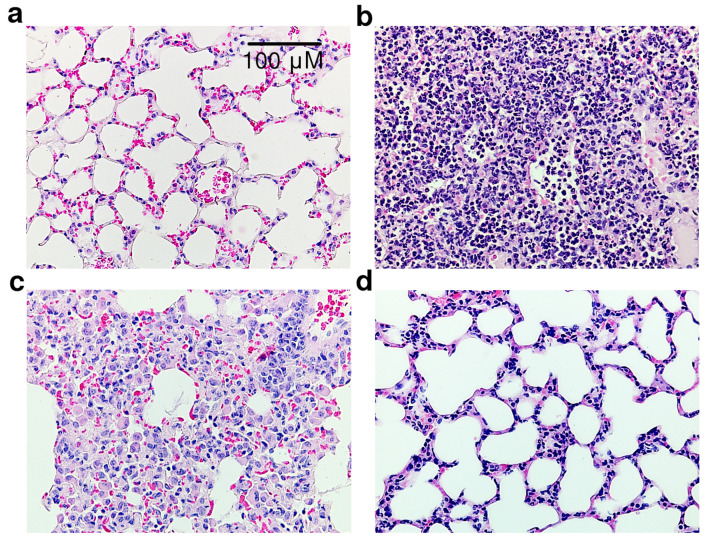
Sections of lung tissue, hematoxylin and eosin staining, ×400 total magnification: (**a**) vehicle; (**b**) LPS + vehicle; (**c**) LPS + dexamethasone; (**d**) LPS + **4a**.

**Figure 9 life-12-01647-f009:**
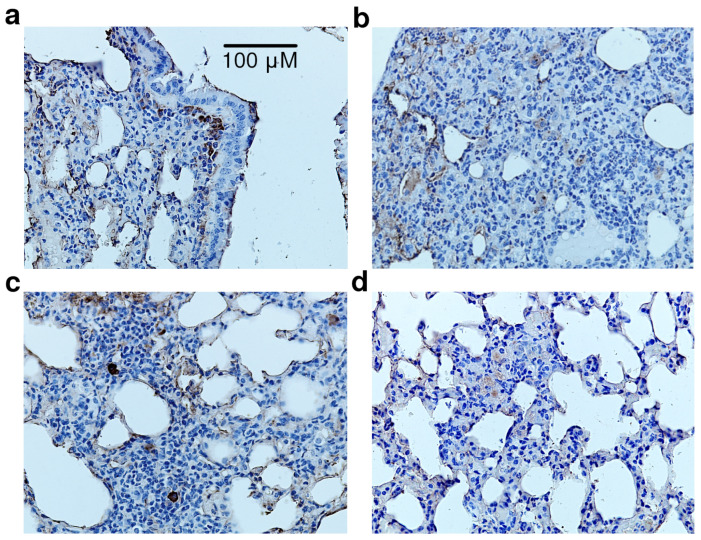
Sections of lung tissue, CD68${}^{+}$ immunostaining, nuclei stained with Mayer’s hematoxylin, ×400 total magnification: (**a**) vehicle; (**b**) LPS + vehicle; (**c**) LPS + dexamethasone; (**d**) LPS + **4a**.

**Table 1 life-12-01647-t001:** Influence of the target quinazoline derivatives on NHE-1, rabbit platelet aggregation, and rat intraocular pressure (mean ± SD).

Compound	NHE-1 Inhibition at (10 nM) (m ± SD, n = 6, %)	NHE-1 IC${}_{50}$ (nM)	Inhibition of Platelet Aggregation at (100 μM) (m ± SD, n = 5, %)	Max IOP Reduction (m ± SD, n = 5, %).
**3a**	37.20 ± 5.97 *	37.2	44.3 ± 14.3	10.08 ± 5.8
**3b**	10.17 ± 3.72 ${}^{\#}$		46.4 ± 9.2 *	24.17 ± 5.05
**4a**	29.37 ± 4.40 *	323.5	39.3 ± 11.9	28.25 ± 7.92
**4b**	16.17 ± 3.75 *${}^{\#}$		47.4 ± 8.9 *	15.45 ± 9.39
Amiloride	5.39 ± 1.82	1230	67.6 ± 1.0 *	18.39 ± 14.58
Acetylsalicylic acid			31.6 ± 4.6 *	

* Statistically significant vs. control (*p* < 0.05, 1-way ANOVA); ^#^ statistically significant vs. reference drug (*p* < 0.05, 1-way ANOVA).

**Table 2 life-12-01647-t002:** Antidepressant activity in tail suspension test.

Compound	Immobilization Time, Mean ± SD, n = 6 (s)
**4a** (4.0 mg/kg)	164.4 ± 23.19 *
**4b** (4.9 mg/kg)	168.2 ± 19.52
Amiloride (2.6 mg/kg)	172.7 ± 8.54 *
Imipramine (8 mg/kg)	112.5 ± 20.56 *
Amitriptyline (10 mg/kg)	50.2 ± 12.71 *
Vehicle	264.8 ± 11.6

* Statistically significant vs. vehicle (*p* < 0.05, Kruskal–Wallis test).

**Table 3 life-12-01647-t003:** Effect of the target quinazoline derivatives on LPS-stimulated C57bl/6j peritoneal macrophages (n = 3).

Compound	NO Synthesis IC${}_{50}\pm$ SE (μM)	IL-6 Secretion IC${}_{50}\pm$ SE (μM)	LDH CC${}_{50}\pm$ SE (μM)
**3a**	>100	>100	>100
**3b**	>100	>100	>100
**4a**	72.96 ± 7.83	51.75 ± 1.65	>100
**4b**	70.98 ± 4.89	64.82 ± 4.31	>100
Dexamethasone	0.003 ± 0.001	0.003 ± 0.001	>100
Amiloride	0.62 ± 0.05	9.52 ± 0.65	>100
Zoniporide	EC${}_{50}$ 15.56 ± 3.76	>100	406.6 ± 27.6

## Data Availability

Not applicable.

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
