# Peer review of "Guanidine Derivatives of Quinazoline-2,4(1*H*,3*H*)-Dione as NHE-1 Inhibitors and Anti-Inflammatory Agents"

_life, 2022, doi:10.3390/life12101647_

Round 1
Reviewer 1 Report
This paper reported the synthesis and evaluation of bis-substituted quinazoline-2,4(1H,3H)-dione guanidine derivatives as NHE-1 inhibitors for the suppression of cytokine-mediated inflammatory response and acute lung injury. This study is based on the authors’ previous report (Scientific Reports 2021, 11, 24380.) which presented the mono-substituted quinazoline-2,4(1H,3H)-dione guanidine derivatives as NHE-1 inhibitors showing anti-inflammatory, antiplatelet, intraocular pressure lowering and antiglycating activity. In structure, the novelty of the new compounds in this manuscript is low, but they are well profiled pharmacologically. The biological results are also interesting and will benefit the research especially about the development of NHE-1 inhibitors for the pharmacological use. In general, the topic fits the scope of the journal, and the manuscript is well-organized, while key issues are required to be addressed before its publication on Life.
1. The NHE-1 and the inflammatory are required to be introduced in the instruction section.
2. The current development of NHE-1 inhibitors (with figure illustration) for suppression of inflammatory and acute lung injury is required to be summarized in the introduction section.
3. For the written form of the C-NMR data, the chemical shifts are generally shown as values with one digit after the decimal point. The authors are suggested to revise them.
4. The times of technical repeats should be noted for the activities (for data in all the tables and figures) of the compounds.
5. The NHE-1 %inhibition activity (3.72±10.17?) of 3b in Table 1 is required to be double-checked or retested, as the standard deviation (SD) is 3 times higher than the tested average value, which is unacceptable.
6. In the table 2, the dosage of 4a, b is required to be noted.
7. In the table 3, the times of technical repeats should be noted, and the values are suggested to be shown with SD values (in the form of xx ± SD).
8. In the figure 2, the meanings of 2x, 3x and 4x* are required to be noted. The same requirement is for the meaning of 4x#.
9. Scale bars are required to be included in Figure 7 and 8.
Author Response
We are grateful for consice review of our paper. Please find the response below.
1. The implication of NHE-1 in inflammation is added to the instruction section.
2. Evidence for NHE-1 inhibitors as anti-inflammatory agents in lung injury is provided in the introduction along with Figure 1.
3. Chemical shifts fort C13-NMR were rounded to one digit after the decimal point.
4. The times of technical repeats is added in all tables and figures.
5. The NHE-1 inhibition value for compound 3b in Table 1 is corrected.
6. Dosage of 4a and 4b is added to the Table 2.
7. Technical repeatys and error values are added to the Table 3.
8. Statistical significance is provided in figure captions.
9. Scale bars are added to Figures 8 and 9.
Reviewer 2 Report
The authors synthesized new derivatives of guanidine derivatives of quinazoline-2,4(1H,3H)-dione and evaluated their NHE-1 inhibitory and anti-inflammatory effects, including in vivo results. The derivatives of quinazoline-2,4(1H,3H)-dione has described their effects in a previous paper by the authors. As discussed in this paper in relation to previous results, this paper contains important results on compound structure of derivatives of quinazoline-2,4(1H,3H)-dione and the effects they produce, and therefore this manuscript contain content appropriate for this journal "Life". However, the authors would like the following points to be considered.
1)In the introduction, the authors do not state why they use quinazoline-2,4(1H,3H)-dione compounds. It may be written in Reference 1, but it is too brief in this paper and that should be described in more detail.
2)The significance of inhibiting NHE-1 should be more detailed in the introduction.
3)The reason for examining Br-substituted compounds is explained as "to balance highly charged guanidine side chains." (line 166-167) But, does introducing bromine into the molecule resolve the charge bias? The authors should explain it in more detail.
Minor errors:
Line 181, table 3 should be table 1.
In table 1, about the number of NHE-1 inhibition of Amiloride, commas should be periods.
line 281, please correct Figure??.
line 329, please check "agents agents"
Author Response
We wnat to thank the reviewer for kind attention to our paper. Please find our response below.
- Rationale behind using quinazoline-2,4(1H,3H)-dione is added to the introduction section.
- Importance of NHE-1 inhibition is provided in the introduction now.
- Reason for bromine introduction to 3b and 4b is added in place.
- Minor errors were fixed, thank you very much for pointing them out
Round 2
Reviewer 1 Report
After the authors’ revision, the quality of this paper was significantly improved, and could reach the required quality standard for Life in my opinion. I suggest accepting it without further revision.
Reviewer 2 Report
The points I raised previously have been appropriately corrected.
Therefore, this article is acceptable.